# Oxidative Stress, Mitochondrial Homeostasis, and Sirtuins in Atrial Fibrillation

**DOI:** 10.3390/ijms27010175

**Published:** 2025-12-23

**Authors:** Jan Krekora, Elzbieta Pawlowska, Marcin Derwich, Jarosław Drożdż, Janusz Blasiak

**Affiliations:** 12nd Department of Cardiology, Medical University of Lodz, 92-213 Lodz, Poland; ejkrekora@op.pl (J.K.); jaroslaw.drozdz@umed.lodz.pl (J.D.); 2Department of Pediatric Dentistry, Medical University of Lodz, 92-217 Lodz, Poland; elzbieta.pawlowska@umed.lodz.pl (E.P.); marcin.derwich@umed.lodz.pl (M.D.); 3Faculty of Medicine, Mazovian University in Plock, 09-240 Plock, Poland

**Keywords:** atrial fibrillation, AF, oxidative stress, mitochondrial quality control, calcium handling, metabolic remodeling, inflammation, fibrosis, calcium handling, mitophagy, sirtuins

## Abstract

Atrial fibrillation (AF) is the most common cardiac arrhythmia. Yet, its treatment has serious challenges and is unsuccessful in a considerable fraction of patients. One reason may be a limited understanding of the molecular mechanisms underlying AF. Recent studies suggest that oxidative stress is involved in AF pathogenesis. Enhanced oxidative stress is largely determined by disrupted mitochondrial homeostasis, as cardiomyocytes heavily rely on mitochondrial energy production and calcium transfer between mitochondria and the sarcoplasmic reticulum. Atrial fibrillation involves metabolic, structural, and electrical remodeling, all of which are influenced by mitochondrial mechanisms. Mitochondrial homeostasis is controlled by mitochondrial quality control (mtQC), which is a multi-pathway mechanism to maintain integrity and functionality of mitochondria. Impaired mtQC may result in disturbed mitochondria-related calcium handling, decreased energy production, mitochondria-related inflammation and fibrosis, and impaired mitophagy. Sirtuins (SIRTs) are a family of seven members of histone deacetylases which have antioxidant properties, and three of them are localized to mitochondria. Therefore, at least some SIRTs may ameliorate enhanced oxidative stress related to damaged mitochondria. SIRTs have shown potential to improve AF outcomes in studies on AF patients and animal models. Therefore, SIRTs may have potential to ameliorate AF by decreasing oxidative stress and restoring mitochondrial homeostasis disrupted in AF. In this narrative review, we provide information on how mitochondrial dysfunctions, expressed as a disturbance in mtQC, contribute to AF through oxidative stress, calcium handling abnormalities, energy deficiency, inflammation and fibrosis, and genetic changes. In addition, we present the protective potential of sirtuins in AF.

## 1. Introduction

Atrial fibrillation (AF) is the most common cardiac arrhythmia, characterized by irregular and disorganized atrial electrical activity that suppresses normal sinus rhythm. AF poses serious health implications for affected individuals and a considerable burden for societies, e.g., new estimates suggest that 10.55 million U.S. adults (about 4.5–5% of the adult population) live with AF [1]. Atrial fibrillation is a major risk of thromboembolic stroke, heart failure, and mortality [2]. Multiple risk factors for AF have been recognized, such as hypertension, obesity, diabetes, heart failure (with both reduced and preserved ejection fraction), obstructive sleep apnea, aging, hyperthyroidism, and heavy alcohol use [3]. The temporal pattern and mode of termination are the primary criteria for AF classification, including first diagnosed AF, paroxysmal AF, persistent AF, longstanding persistent AF, and permanent AF [4]. The diagnosis of AF is based on the absence of P waves and irregular R-R interval on a 12-lead ECG [5]. The 2020 European Society of Cardiology guidelines recommend that a rhythm strip not shorter than 30 s can also indicate AF [4]. Contemporary AF management is based on three pillars, integrated pathways, denoted the AF Better Care (ABC) [6]. They are symptom-directed decisions on rate versus rhythm control, optimizing stroke prevention, and managing risk factors for cardiovascular and other comorbidities.

Despite its commonness and serious health consequences, an optimal strategy for AF management remains a challenge [7]. This is at least partly due to insufficient knowledge of the molecular mechanisms behind AF pathogenesis. Recent studies suggest the role of oxidative stress in AF pathology [8]. The stress is associated with an overproduction of reactive oxygen and nitrogen species (RONS), which may damage cellular macromolecules, including nucleic acids, proteins, and lipids. Oxidative DNA damage has recently been shown to be linked to AF [7,9]. However, as with most cases of the association between oxidative stress and pathological phenotype, it is difficult to definitively determine whether the stress is a cause or a result of the disease. Despite this uncertainty, it is clear that higher levels of RONS in AF originate from impaired mitochondria, which generate them even in normal conditions, and mitochondrial impairment leads to RONS overproduction. Therefore, managing mitochondrial homeostasis may be an important element in AF pathophysiology, especially since mitochondria play a critical role in maintaining the electrophysiological stability and function of the heart [10]. The involvement of impaired mitochondrial function in AF pathogenesis was confirmed in several studies (reviewed in [11]).

Due to an important role in oxidative stress, inflammation, aging, and metabolism, mitochondrial homeostasis is important in preventing not only cardiovascular syndromes but also many other diseases [12]. Therefore, maintaining mitochondrial homeostasis is crucial for maintaining the organism’s overall homeostasis. Mitochondrial quality control (mtQC) is a set of mechanisms, including mitochondrial biogenesis, mitochondrial dynamics, mitophagy and mitochondrial DNA damage response to ensure mitochondrial fitness and functions (reviewed in [13]). All these mechanisms contain many pathways, making mtQC a complex and sophisticated organism’s reaction to keep its homeostasis.

In this narrative review, we provide information on how mitochondrial dysfunctions, expressed as disturbances in mtQC, contribute to AF through oxidative stress, calcium handling abnormalities, energy deficiency, inflammation, fibrosis, and genetic changes. In addition, we presented the protective potential of sirtuins in AF. This association of mtQC with sirtuins is justified by the antioxidant properties and mitochondrial localization of some of them. Our search strategy concentrated on mechanisms, clinical implications, biomarkers, and therapeutic approaches in AF, emphasizing oxidative stress, mitochondrial homeostasis, and sirtuins. It was based on publications from PubMed, Embase, Google Scholar, ScienceDirect, and Cochrane Library. We used search strings including “atrial fibrillation” and one of the following terms: “oxidative stress”, “ROS”, “mitochondria”, or “sirtuin.” Publications from the past decade were prioritized, unless foundational studies were necessary. The species included were humans and relevant animal models. All article types, including original research, reviews, and meta-analyses, were considered, with no language restrictions.

## 2. Oxidative Stress in Atrial Fibrillation

Oxidative stress is linked to many diseases, but in most cases, the causal connection between stress and a specific syndrome remains unclear. Extensive details on the role of oxidative stress in AF development have been outlined in a recent review by Pfenniger et al. [8].

Reactive oxygen and nitrogen species are produced in the leaking mitochondrial electron transport chain (mtETC) and after the activation of various enzymes, including NADPH oxidases (NOX), xanthine oxidase, and uncoupled nitric oxide synthase (NOS) [14]. Cardiomyocytes are rich in mitochondria and, therefore, may generate a high amount of RONS and may be considered the main source of RONS in the myocardium [8]. Elevated level of RONS in the myocardium was observed in several heart syndromes, including heart infarction, heart failure, and AF [15]. Also, NOX, xanthine oxidase, and NOS are expressed in the myocardium (reviewed in [8]).

It should be emphasized that most data on the role of oxidative stress in AF patients are obtained indirectly, as atrial tissue can only be obtained in postoperative AF cases, not in paroxysmal or persistent AF [16]. Using a goat model of pacing-induced AF, it was demonstrated that the upregulation of atrial NADPH oxidases is an early but temporary event in the progression of AF [17]. Additionally, the study indicated that changes in the sources of RONS with atrial remodeling could explain why statins are effective in primary prevention of AF but not in its treatment. It was demonstrated that NOX2 was upregulated only in patients with paroxysmal/persistent AF, but not with permanent AF, and was responsible for the overproduction of isoprostanes [18]. AF was identified early in community-acquired pneumonia and was linked to NOX2 activation [19]. A study showed that atrial perfusion with hydrogen peroxide in rats caused atrial fibrillation [20,21]. However, that research involved acute pharmacological interventions that might not reflect the chronic pathophysiology of AF.

Several AF risk factors, including heart failure and ischemic/nonischemic cardiomyopathy, are associated with elevated oxidative stress [8]. The stress is induced by electron leakage from mitochondrial complexes I and III and by reduced antioxidant defense. Also, diabetes is associated with elevated RONS production due to hyperglycemia and mitochondrial dysfunction [22]. It was demonstrated that mitochondria in atrial tissue from type 2 diabetic patients showed diminished capacity for glutamate- and fatty acid-supported respiration, in addition to increased myocardial triglyceride content [23]. Moreover, those patients exhibited increased mitochondrial production of hydrogen peroxide during oxidation of carbohydrate- and lipid-based substrates, depleted glutathione levels, and evidence of persistent oxidative stress in their atrial tissue.

Sustained or recurrent episodes of AF can cause long-term changes in the electrical properties of the atria, known as electrical remodeling, that increase the atria’s susceptibility to arrhythmias [24]. The key features of electrical remodeling in AF are shortening of the atrial action potential duration, changes in potassium and calcium channel function, and increased vulnerability to reentry circuits. Several ionic currents are redox-dependent in AF, as RONS produced by NOX2 and mitochondria regulate them. These include the L-type Ca^2+^, inward-rectifier K^+^ current, and constitutively active form of acetylcholine-dependent K^+^ current (reviewed in [8]).

Catheter ablation has emerged as a minimally invasive procedure to treat AF, particularly in patients with paroxysmal or persistent AF, especially those who are refractory or intolerant to antiarrhythmic drugs [25]. Electrical remodeling contributes to the progression from paroxysmal to persistent AF and may decrease the efficacy of drug therapy and ablation in AF.

Besides electrical remodeling, structural remodeling also occurs in AF, especially during its later stages, probably due to prolonged exposure to risk factors [26]. It involves physical and anatomical changes in the atrial tissue that occur during AF. They contribute to the persistence and worsening of arrhythmia and can make treatment more difficult. The key features of structural remodeling are atrial fibrosis, atrial dilation, myocyte degeneration and dedifferentiation, inflammation and oxidative stress, calcium overload and stretch-activated pathways. These features may impede the propagation of electrical impulses through the atria, leading to heterogeneity of electrical conduction [27]. Fibrosis is a crucial factor in structural remodeling in AF and plays a significant role in AF treatment, especially with ablation. An involvement of NOX2 in cardiac fibrosis mediated by angiotensin II (ANGII) was shown in mice [28]. Apart from NOX2, NOX4 may also be involved in atrial fibrosis in AF, and it has been shown that this involvement in human cardiac fibroblasts might be mediated by the transforming growth factor beta 1 (TGFB1)-NOX4 signaling pathway [29]. Several studies involving humans and various animal models, including serine/threonine-protein kinase STK11 (LKB1) knockout mice and mice overexpressing catalase, an enzyme that decomposes hydrogen peroxide, demonstrated the involvement of mitochondrial RONS in structural remodeling in AF [30,31,32].

Paired like homeodomain 2 (PITX2) belongs to the RIEG/PITX homeobox family and acts as a transcription factor that triggers an antioxidant response to support cardiac repair [33]. Mice deficient in the *PITX2* gene showed an increased AF susceptibility [34]. However, the molecular mechanism underlying this effect remains debated. It has been shown that *PITX1*(+/−) mice exhibit longer P-wave duration, reflecting slower atrial conduction, as well as a greater inducible AF burden and persistent AF compared with wild-type mice. These issues were mitigated by 2-hydroxybenzylamine, a natural antioxidant compound [35]. In mutant mice, RONS-related protein adducts increased in the atria, coinciding with lower levels of antioxidant gene expression. The atria of *PITX1*(+/−) animals exhibited impaired mitochondrial function, characterized by disrupted mitochondrial integrity and reduced biogenesis. These issues were somewhat alleviated or avoided by 2-hydroxybenzylamine. The findings emphasize the role of lipid dicarbonyl mediators of oxidative stress in proarrhythmic remodeling and the susceptibility to AF linked to PITX2 deficiency. This suggests opportunities for creating genotype-specific treatments to prevent AF.

Increased RONS levels, originating from NOX isoforms or mitochondria, may promote AF through abnormal Ca^2+^ handling in atrial myocytes, which display higher expression of the sarcoplasmic reticulum calcium ATPase 2A (SERCA2A), and a lower expression of phospholamban, a SERCA2A regulator [36,37,38]. Consequently, atrial myocytes display higher Ca^2+^ content than their ventricular counterparts.

This strong evidence for oxidative stress’s role in AF pathogenesis supports the use of antioxidant therapies to prevent and treat AF. As presented by Pfenniger et al., these therapies can target RONS directly, their sources of production, and signaling pathways that promote an AF arrhythmogenic substrate [8]. Although systematic antioxidant therapies did not yield satisfactory results, novel atrial-specific gene-based therapies demonstrated promising effects in large-animal AF models [39,40]. Further information on targeting oxidative stress in AF will be provided in the subsequent sections.

Several antioxidant molecules were experimentally tested in AF. Resveratrol and dapagliflozin are described in a separate section. It was shown that inhaled H_2_ lowered atrial RONS, inhibited NOX4 activation, suppressed the NLRP3 inflammasome and fibrosis, thereby reducing AF inducibility in an ANGII-induced rat AF model [41]. In rat AF models and clinical studies, honey-fried licorice decoction, a traditional preparation used in Chinese herbal medicine, downregulated NOX2 expression and RONS in atrial tissue, restored antioxidant enzyme balance (SOD, CAT, GSH/GSSG), reduced TGF β1/SMAD3 signaling, and decreased AF susceptibility [42]. Apocynin, a NOX inhibitor, was proposed to inhibit NADPH oxidase activity, especially NOX2, lowering RONS production in atrial tissue. While highlighted in hypothesis papers for AF management, direct experimental validation in AF models was observed [37,43]. Statins inhibit NOX2-NADPH oxidase activity and reduce postoperative AF through antioxidant effects. They show limited benefit in long-standing AF due to alternative sources of ROS [17]. Clinical and experimental studies report that vitamins C and E reduce RONS, lower AF recurrence after cardioversion, and decrease postoperative AF incidence [44]. In cardiovascular disease contexts, CoQ10, flavonoids, and carotenoids reduce oxidative stress. Although not AF-specific, they are promising antioxidant candidates [45]. Highlighted in arrhythmia-focused reviews. MitoTEMPO targets mitochondrial ROS, potentially attenuating oxidative remodeling in AF [46]. Nitric oxide synthase inhibitors are mentioned as a promising antioxidant strategy, with potential to modulate ROS sources in AF [47]. These and other molecules are presented in Table 1.

In summary, several molecular mechanisms may underlie oxidative stress’s role in AF pathogenesis. Enhanced RONS levels in AF originate from mitochondria-rich atrial cardiomyocytes, which may generate RONS due to impaired mtETC, activation of some oxidases, and compromised mtQC (Figure 1). Therefore, these mechanisms can be considered in therapeutic strategies for AF.

## 3. Mitochondrial Homeostasis in Atrial Fibrillation

Although a significant portion of the results presented in the previous section can be directly or indirectly linked to dysfunctional mitochondria and, consequently, impaired mtQC, the focus was on oxidative stress without exploring the mechanisms underlying its effects.

As mentioned, many pathways contribute to mtQC, including a multilayered regulatory network that is essential for maintaining mitochondrial integrity and functional homeostasis (reviewed in [13]). This system coordinates its key components: mitochondrial dynamics—fission and fusion; mitophagy—a specialized form of autophagy that selectively degrades damaged or no longer needed mitochondria; biogenesis; calcium homeostasis; proteostasis; unfolded protein response; import machinery; and DNA damage response (DDR) (Figure 2). Some aspects of mtQC overlap; for example, mitochondrial DDR, unlike its nuclear counterpart, may trigger the degradation of mitochondria if they contain DNA damage beyond the organelle’s repair capacity [49].

Mitochondrial quality control is essential for providing energy to the cardiovascular system (reviewed in [10]). Dysfunctional mtQC was implicated in many cardiovascular diseases, including ischemia–reperfusion, atherosclerosis, heart failure, cardiac hypertrophy, hypertension, diabetic and genetic cardiomyopathies, and Kawasaki Disease [10]. Consequently, mtQC is essential to promote cardiovascular health. Some aspects of mtQC were reported to be impaired in AF, suggesting potential avenues for AF prevention and treatment.

### 3.1. Mitochondria-Related Calcium Handling

Abnormal calcium (Ca^2+^) handling in mitochondria plays a key role in AF pathogenesis (reviewed in [50]). It interferes with the balance among energy generation, oxidative stress control, and intracellular signaling in atrial heart cells. In normal conditions mitochondria take up Ca^2+^ during increased workload to stimulate the tricarboxylic acid (TCA) cycle, enhancing ATP production and maintaining redox balance. In AF mitochondrial Ca^2+^ accumulation is decreased, especially during stress or increased workload [51]. This leads to decreased ATP production, impaired regeneration of reduced nicotinamide adenine dinucleotide (NADH) and reduced flavin adenine dinucleotide (FADH_2_), essential for oxidative phosphorylation. Insufficient reduced nicotinamide adenine dinucleotide phosphate (NADPH), which is involved in the antioxidant defense through several pathways, including glutathione regeneration, increases oxidative stress and disturbs anabolic reactions [52].

Calcium transfer between mitochondria and the sarcoplasmic reticulum (SR) is essential for cellular functions, particularly in excitable cells, including cardiomyocytes [53]. The proximity between SR and mitochondria facilitates this transfer. In AF, this spatial organization is disrupted, mainly due to microtubule destabilization, resulting in inefficient Ca^2+^ transfer, increased proarrhythmic Ca^2+^ sparks, and changes in excitation–contraction coupling [54]. Impaired mitochondrial Ca^2+^ handling adds to spontaneous Ca^2+^ release from the SR, electrical instability, and promotes reentry circuits, a hallmark of AF pathophysiology [50,55,56,57].

A recent study on AF patients’ derived material confirmed the importance of impaired atrial mitochondrial calcium handling in AF pathogenesis [58]. The study revealed that during increased workload, mitochondrial Ca^2+^ buildup was reduced in AF, linked to impaired NADH and FADH_2_ regeneration. Disorganization of the SR and mitochondria, along with microtubule destabilization, was detected. These results were validated in human iPSC-derived myocytes, where nocodazole treatment displaced mitochondria and increased proarrhythmic Ca^2+^ sparks, which MitoTEMPO could rescue. Ezetimibe also lessened arrhythmogenic Ca^2+^ release events in AF myocytes and nocodazole-treated human iPSC-derived cardiac myocytes. A retrospective patient review found a lower AF burden among those taking ezetimibe. The authors concluded that mitochondrial Ca^2+^ uptake was impaired in AF atrial myocytes, likely due to disrupted SR-mitochondria spatial organization caused by destabilized microtubules. Enhancing mitochondrial Ca^2+^ uptake might serve as a protective strategy against arrhythmogenic events.

These findings suggest that focusing on mitochondrial Ca^2+^ regulation may offer a new therapeutic strategy for AF, particularly in persistent or resistant cases [59]. In therapeutic strategies, modulation of mitochondrial Ca^2+^ transporters were considered. They were mitochondrial calcium uniporter (MCU), sodium-calcium exchanger (NCLX), and ryanodine receptor 2 (RyR2) [60,61,62].

Ezetimibe, a lipid-lowering drug, demonstrated promise in AF, especially when combined with statins [63,64]. Although a detailed mechanism underlying the observed profitable effects of ezetimibe in AF is not known, its influence on calcium homeostasis may play a role, as it increases mitochondrial Ca^2+^ uptake and decreases arrhythmogenic Ca^2+^ release [65,66].

In summary, cardiomyocytes heavily rely on calcium transfer between mitochondria and the sarcoplasmic reticulum, which depends on the close physical proximity of these two organelles, a connection disrupted in AF. Reduced mitochondrial Ca^2+^ accumulation in AF might be connected to impaired regeneration of NADH and FADH_2_. Mitochondrial regulation of calcium homeostasis could be considered a therapeutic target in AF.

### 3.2. Energy Production

The three-way link between energy production, mitochondrial health, and atrial fibrillation (AF) is increasingly recognized as a crucial aspect of AF’s underlying mechanisms. As the main source of energy production is oxidative phosphorylation in mitochondria, much of the mechanisms leading to disturbed energy production associated with AF is described in the remaining subsections of this section.

Cardiac myocytes need large amounts of ATP for excitation–contraction coupling. The heart’s energy demand in AF is greater than in a normal heart because atrial cells in AF fire quickly and chaotically, often exceeding 300–500 impulses per minute [67]. This causes ongoing depolarization and repolarization, significantly increasing the strain on the ion pumps, which require a substantial amount of ATP to restore the ionic gradient. Inefficient mechanical contraction in AF due to losing the atrium’s coordinated contraction causes a conflict with muscle cells and wastes energy that must still be supplied [68].

Oxidative phosphorylation, essential for energy production in heart tissue, decreases in AF, as confirmed by multiple studies involving human atrial tissue and animal models. Proteomics and metabolomics analyses of left atrial biopsies from AF patients showed reduced levels of mitochondrial enzymes necessary for OXPHOS, such as GPD2 and components of mtEC [48]. Gene set enrichment analysis revealed significant decreases in oxidative phosphorylation pathways and ATP biosynthesis processes in AF atria compared to controls. These alterations are associated with diminished mitochondrial energy metabolism and structural remodeling.

The inducibility of trial tachyarrhythmia, as well as interatrial conduction time and fibrosis, was higher in rats with diabetes mellitus in functional measurements of mitochondrial respiration [48]. Diabetic rats generated higher levels of mitochondrial-RONS and showed reduced complex I-linked oxidative phosphorylation capacity. Treatment with empagliflozin, a sodium-glucose co-transporter-2 inhibitor, improved mitochondrial function. Consequently, diabetic rats with atrial arrhythmia produce less energy because of impaired OXPHOS and other mitochondrial dysfunctions.

Conflicting data relate atrial pacing to the risk of AF, and the impact of atrial pacing on mitochondrial function remains poorly understood [69]. It was demonstrated that atrial pacing from cardiovascular implantable electronic devices (CIEDs) exceeding 50% is associated with higher levels of mitochondrial spare respiratory capacity and coupling efficiency [70]. These results demonstrate that atrial pacing improves mitochondrial performance in PBMCs and enhances left ventricular contractile function in patients with CIEDs, supporting observations of the protective effect of atrial pacing against atrial arrhythmia. Early AF episodes can lead to temporary depletion of phosphocreatine and ATP, but with chronic AF, there is often some recovery of energy balance, indicating adaptive remodeling [71,72].

As mentioned in the previous section, SR-mitochondrial calcium cycling is disrupted in AF, resulting in elevated spontaneous Ca^2+^ release events and elevated levels of cytosolic Ca^2+^, which must be actively transported back into SR or out of the cell [73]. These processes also require a significant amount of ATP.

Enhanced ATP production imposes stress on mitochondria associated with RONS overproduction, which are produced even in normal conditions. These RONS damage the mitochondrial membranes and proteins and impair the functioning of mtETC, resulting in further RONS production (“vicious cycle”).

In summary, high ATP turnover in AF leads to effects that exacerbate mitochondrial dysfunctions, resulting in energy deficits in the atrial tissue cardiomyocytes and are serious challenges for mtQC.

### 3.3. Inflammation and Fibrosis

Mitochondrial damage can activate inflammatory pathways and promote fibrotic remodeling, both of which contribute to AF [74]. Upon damage, mitochondria release molecules collectively referred to as mitochondrial-derived damage-associated molecular patterns (DAMPs), containing molecules that are important in inflammatory and immune responses in various diseases (reviewed [75]). These include mtDNA, cardiolipin, and RONS. As mentioned, RONS can promote and/or enhance oxidative stress, and mtDNA can activate toll-like receptor 9 (TLR9) and cyclic GMP–AMP synthase–stimulator of interferon genes (cGAS-STING) pathways [76]. cGAS detects cytosolic DNA, including mtDNA, and activates STING, which stimulates the production of type I interferons and other inflammatory cytokines. It was shown that mitochondrial damage mediated STING activation driving obesity-mediated AF in rats [77]. However, it is not entirely clear whether the cGAS-STING pathway is a friend or foe in cardiac diseases; it may play a role in the low-grade inflammation associated with AF [78].

Calprotectin is a heterodimer of two proteins from the S100 family, S100A8 and S100A9, which play a role in the inflammatory response (reviewed in [79]). Recent studies in patients and mice with AF have shown that S100A8/A9 from macrophages promotes atrial remodeling and AF through the TLR4/NFKB signaling pathway [80]. Calprotectin may connect oxidative and inflammatory pathways in cardiovascular diseases, including arrhythmias (reviewed in [81]).

Under normal conditions, cardiolipin binds to the inner mitochondrial membrane. Still, during mitochondrial damage or stress, it can translocate to the outer mitochondrial membrane and directly bind to the NOD-, LRR-, and pyrin domain-containing protein 3 (NLRP3) inflammasome. This cytosolic pattern recognition receptor plays a central role in the innate immune system (reviewed [82]). Its activation results in the cleavage of pro-caspase-1 into active caspase-1, which then processes pro-IL-1β and pro-IL-18 into their active, pro-inflammatory forms. Increased activity of the NLRP3 inflammasome was observed in the atrial cardiomyocytes of patients with paroxysmal AF and chronic AF [83]. CM-specific knock-in mice expressing constitutively active NLRP3 (CM-KI mice) developed spontaneous premature atrial contractions and inducible AF, which was reduced by a specific NLRP3 inflammasome inhibitor, MCC950. These mice exhibited ectopic activity, abnormal Ca^2+^ release from SR, shortened atrial effective refractory period, and atrial hypertrophy. Knocking down the NLRP3 gene suppressed AF development in CM-KI mice. Ultimately, genetic inhibition of NLRP3 prevented AF in a mouse model of spontaneous AF.

A study involving patients undergoing cardiac surgery with paroxysmal or long-standing persistent AF confirmed that NLRP3 was active in human atrial cardiomyocytes, supporting its role in AF pathophysiology [83]. This study was conducted on 54 patients, analyzing biopsies from the right atrial appendage and ventricular tissue. Another study examining atrial biopsies taken during surgery from AF patients with a recent history of cigarette smoking showed that smoking was associated with increased ER stress and NLRP3 inflammasome activation in atrial tissue [84]. That research, which involved 44 AF patients, was supplemented with additional studies on iPSC-derived human atrial cardiomyocytes.

It should be recognized that there are significant limitations when extrapolating from mouse models to humans. Mouse NLRP3 regulation and diversity differ from humans—for example, NLRP1 paralogs and overexpression or knockout in transgenic mice may not accurately reflect human variability in gene regulation and immune responses. Mouse models often involve targeted genetic manipulation, whereas in humans, factors such as polymorphisms, epigenetics, aging, lifestyle, and comorbidities influence outcomes. Differences in atrial size, electrophysiology, heart rate, and immune system function between mice and humans can affect the application of findings to AF pathogenesis. Lastly, mouse studies generally focus on pacing-induced AF, which mimics acute arrhythmogenic triggers rather than the chronic remodeling observed in human persistent AF.

The interaction between the NLRP3 inflammasome and gut microbiota might be causal for AF [85]. It should also be emphasized that the atrial-specific NLRP3 inflammasome is a key causal factor in the development, progression, and recurrence of AF after ablation [86].

Because inflammation may induce mitochondrial damage through mtDNA lesions, disrupting mitochondrial dynamics and impairing mitophagy, if this inflammation results from mitochondrial damage, a feedback loop is established between mitochondria and inflammation. Again, a mitochondrial “vicious cycle” may occur in AF pathogenesis.

Atrial fibrosis is characterized by a metabolic reprogramming in atrial fibroblasts and cardiomyocytes. This metabolic dysregulation involves disturbances in fatty acid oxidation, glucose metabolism, and amino acid metabolism, which are closely connected to mitochondria [41]. A 3D organoid model derived from human-induced pluripotent stem cell-derived cardiomyocytes was used to study mitochondrial impairment and cardiac fibrosis in AF [87]. Rapid pacing at 3 Hz resulted in reduced peak contraction amplitude and contraction speed. Multi-omics analysis revealed mitochondrial damage, with notable alterations in succinate dehydrogenase subunits (SDHA–D) expression and the peroxisome proliferator-activated receptor gamma coactivator 1-alpha *PPARGC1A* gene, which encodes PGC-1α—a crucial regulator of mitochondrial biogenesis. Cytosolic cytochrome-*c* levels, indicating mitochondrial integrity, increased. Evidence of profibrotic signaling activation included the upregulation of the type-1 angiotensin II receptor (AGTR1), which is associated with fibrosis and remodeling of cardiac and vascular tissues, along with increased TGFB1. These changes elevated collagen-I and alpha-smooth muscle actin (α-SMA), markers of myofibroblast differentiation characteristic of fibrosis. The organoids’ transcriptional profile closely resembled human atrial fibrosis signatures, confirming this model’s relevance for exploring mitochondrial-fibrotic remodeling in early atrial fibrillation.

In summary, damage to mitochondria may trigger inflammation and promote fibrotic remodeling that could contribute to AF development. The NLRP3 inflammasome might be an important mechanistic link between mitochondrial damage-related inflammation and AF. The cGAS-STING pathway, which detects mtDNA from damaged mitochondria, may also be involved in the low-grade inflammation associated with AF. However, inflammation can cause mtDNA damage, leading to a mutually reinforcing cycle (“vicious cycle”) in AF development.

### 3.4. Mitophagy

Mitophagy is a specialized type of macroautophagy that targets damaged mitochondria. When mitochondrial damage occurs, the inner mitochondrial membrane depolarizes, and the import of PTEN-induced putative kinase 1 (PINK1) into the mitochondria, along with its degradation by presenilin-associated rhomboid-like protein (PARL), is suppressed. This results in increased accumulation of PINK1 on the outer mitochondrial membrane [88]. Next, E3 ubiquitin-protein ligase parkin (PARKIN) is recruited from the cytosol. PINK1 phosphorylates PARKIN, which then ubiquitinates many downstream autophagosome-related proteins and promotes the local formation of autophagosomes with microtubule-associated proteins 1A/1B light chain 3B (LC3) and optineurin (OPTN). Mitophagy can also occur through a pathway that is unrelated to PINK1/PARKIN [89].

Impaired mitophagy leads to ineffective removal of damaged mitochondria, resulting in the accumulation and ongoing release of DAMPs, which cause inflammation typical of AF and other heart diseases [90]. Compromised mitophagy may also contribute to AF pathogenesis in other pathways.

An increased number and size of mitochondria were observed in atrial myocytes of AF patients [91]. A decreased expression of LC3B II, the membrane-bound, lipidated form of LC3 associated with autophagosomes, was also noted in AF patients along with a decrease in the LC3B II/LC3B I (cytosolic, non-lipidated form of LC3) ratio, indicating decreased autophagosome formation. Altogether, these effects suggest mitophagy impairment in AF patients, which may be underscored by dysfunctions in the process of delivering mitochondria into autophagosomes.

A total of 444 differentially expressed genes in rats with AF were identified, including AF-related mitophagy ion channel genes: BCL2 associated X, apoptosis regulator (*BAX*), catenin beta 1 (*CTNNB1*), dihydropyrimidinase like 2 (*DPYSL2*), epoxide hydrolase 1 (*EPHX1*), glutamate-ammonia ligase *(GLUL*), G protein subunit beta 2 (*GNB2*), macrophage migration inhibitory factor (*MIF*), MYC Proto-Oncogene, BHLH transcription factor (*MYC*), and *TLR4* [92]. Four hub genes—*BAX*, *GLUL*, *MIF*, and *TLR4*—were identified. In vivo studies revealed a disorganized myocardial cell structure, abnormal collagen fiber growth, widened interstitial spaces, the development of fibrous septa, and uneven cytoplasmic staining. In summary, *BAX*, *MIF*, and *TLR4* are crucial genes linking mitophagy and ion channels in AF, likely impacting the immune microenvironment through modulation of immune cell infiltration.

Mitogen-activated protein kinase 14 (MAPK14, p38αMAPK)) is a type of osmotic protein kinase that is stimulated by various forms of cellular stress [93]. It can change the mitochondrial membrane potential, RONS, and Ca^2+^ homeostasis. Studies on rats and HL-1 cells treated with angiotensin II (ANG II), a profibrogenic factor, revealed the importance of mitophagy in AF [94]. The rat model showed that inhibition of MAPK14 downregulated the AIF Family Member 2 (*AIFM2*) gene encoding an oxidoreductase that contributes to mitochondrial-related apoptosis [95]. This effect was associated with an improvement in electrical atrial conduction. Both AF models, in vivo and in vitro, demonstrated that the MAPK14/AIFM2 pathway enhanced ANG II-induced AF by regulating mitophagy-related apoptosis. These results can be generalized, assuming that mitochondrial RONS can induce MAPK14 and AIFM2, resulting in mitophagy and apoptosis, which, in turn, lead to atrial structural remodeling, ultimately resulting in AF. Therefore, inhibition of the MAPK14/AIFM pathway may ameliorate AF induced by ANG II by regulation of mitophagy-related apoptosis.

As mentioned in the previous sections, catheter ablation is an emerging strategy for AF treatment. However, in several cases, AF recurrence is observed. The mechanism(s) behind such effects are not completely clear, but it is hypothesized that very late-onset (>1 year) AF recurrence is underlined by a different mechanism than AF relapse during the first year after ablation [96]. A serum PARKIN level below the median was independently associated with very late-onset AF recurrence, but not within the first year after the treatment [97]. In contrast, ATG5, a marker of bulk autophagy, was not associated with AF recurrence. Therefore, impaired mitochondrial autophagy could play a role in the development of long-term AF recurrence.

In summary, mitophagy is emerging as a key mediator in the development and progression of AF, as impaired mitophagy fails to remove damaged mitochondria, leading to the accumulation and release of DAMPs that cause inflammation typical of AF. Impairment in mitophagy in AF is often linked to defective autophagosome formation. Many differentially expressed genes were identified in a rat model of AF. Additionally, mitophagy-related apoptosis may contribute to AF. It is worth noting that impaired mitophagy may also play a role in AF recurrence after ablation and could serve as an important prognostic marker in this critical procedure.

## 4. Sirtuins in Atrial Fibrillation

Sirtuins (silent information regulators, SIRTs) are an important element of epigenetic regulation of gene expression, as they are class III NAD+-dependent histone deacetylases. Humans have 7 SIRTs (SIRT1-7) localized in various cellular components, including the nucleolus, cytoplasm, and mitochondria [98]. Emerging evidence shows an important role of sirtuins in the development of various syndromes, including cardiovascular disease [99]. Various mechanisms can underline both the physiological and pathological functions of SIRTs beyond their regulation of gene expression. These include antioxidant and redox signaling, energy metabolism, DNA repair, and mtQC, including mitophagy [100,101,102]. Therefore, SIRTs display activities that can be important in oxidative stress-related AF pathogenesis. However, the antioxidant properties of SIRTs should not be overgeneralized, as SIRT4 overexpression was recently found to accelerate heart failure development under pressure overload, primarily by enhancing profibrotic transcriptional signaling through RONS-driven mechanisms [103].

While many sirtuins, including SIRT1 and SIRT3, activate antioxidant defenses and enhance mitochondrial function, SIRT4’s role depends on the context: in mild stress, it may help regulate metabolism, but under severe pressure overload, its signaling promotes ROS-driven profibrotic remodeling, worsening heart failure. Therefore, sirtuins are not uniformly protective; their effects vary based on isoform-specific enzymatic activity, cellular environment, and stress level. SIRT4’s unique metabolic and signaling functions can shift the balance toward maladaptation during chronic overload.

In their primary function in mitochondria, SIRTs, mainly SIRT3, but also SIRT4 and SIRT5 to some extent, remove acetyl groups from lysine residues on mitochondrial proteins [104]. SIRT3 deacetylates and activates enzymes involved in oxidative phosphorylation and the TCA cycle. These include (1) subunits of complex I and II in the mtETC, which enhance mtETC efficiency; (2) components of ATP synthase, leading to increased ATP production; and (3) isocitrate dehydrogenase 2 and long-chain acyl-CoA dehydrogenase, which optimize substrate oxidation [105,106]. Therefore, when SIRTs are active, they increase mtETC flux, strengthen the proton gradient, and enhance ATP synthases, responding to higher ATP production during energy-demanding conditions in AF.

As mentioned, PGC-1α is a crucial protein to regulate mitochondrial biogenesis and energy metabolism and several studies showed the involvement of sirtuins, in particular SIRT1, in its control [107]. Fenofibrate is a PPAR-α agonist and SIRT1 regulator and it was shown that it rescued lipotoxic cardiomyopathy [108]. Using left atria derived from AF patients, a rabbit model of AF, and HL1 cells, it was demonstrated that fenofibrate inhibited atrial metabolic remodeling in AF through the peroxisome proliferator-activated receptor alpha (PPAR-α)/SIRT1/PGC-1α pathway [109]. A subsequent study, using the same models as the earlier work and employing honokiol, a SIRT3 agonist, showed that SIRT3 was downregulated in AF patients and the rabbit/HL1 cell model [110]. This led to abnormal expression of its downstream metabolic key factors, which were restored by honokiol. Therefore, these two studies demonstrate that SIRT1 and SIRT3 may play a role in AF development by their involvement in atrial metabolic remodeling.

As mentioned, atrial fibrosis is a key factor in the development of AF. The fibrogenic processes in the atrium are regulated by several factors, including the transforming growth factor beta 1 (TGFB1)-SMAD pathway [111]. A reduced expression of SIRT1 was observed in the right atrial appendage tissues of AF patients with a concomitant increase in the degree of fibrosis [112]. In atrial fibroblasts, activating SIRT1 can inhibit the expression of the TGFB1-SMAD pathway and decrease fibrosis development, while inhibiting SIRT1 reduces its suppressive effect on that pathway. Therefore, SIRT1 may prevent atrial fibrosis by downregulating the TGFB1-SMAD pathway, and this mechanism can be considered in the AF prevention and treatment.

Aging is a risk factor for the onset of AF, and SIRTs expression declines with age (reviewed in [113,114]). To investigate the role of SIRTs in age-related AF and identify the underlying molecular mechanisms, levels of SIRT1-7 in the atria of individuals divided into age groups and aging rats [115]. The mRNA levels of SIRT1 and SIRT5 were lower in the atria of elderly patients than in those of their younger counterparts; however, the protein levels of SIRT1 decreased, while those of SIRT5 remained unchanged. To confirm the role of SIRT1 in age-related AF, mice were genetically engineered to specifically knock out SIRT1 in the atria and right ventricle, resulting in increased atrial diameter and greater susceptibility to AF. SIRT1 deficiency activated atrial necroptosis by increasing acetylation of receptor-interacting serine/threonine kinase 1 (RIPK1) and subsequent phosphorylation of mixed lineage kinase domain-like protein (MLKL). Furthermore, necrostatin-1, a necroptosis inhibitor, reduced atrial necroptosis, decreased atrial diameter, and lessened AF susceptibility in SIRT1-deficient mice. Resveratrol prevented age-related AF in rats by activating atrial SIRT1 and inhibiting necroptosis. Therefore, SIRT1 helps counteract age-related AF by suppressing atrial necroptosis through regulation of RIPK1 acetylation, and therefore, activating SIRT1 or inhibiting necroptosis could serve as potential therapeutic strategies for age-related AF.

Apart from aging, diabetes is a considerable risk factor for AF (reviewed in [116]). Inhibitors of sodium-glucose cotransporter-2 (SGLT2), an important player in diabetes pathogenesis, were reported to be helpful in treating heart failure and reducing AF risk (reviewed in [117,118]. SGLT2i were shown to interact with SIRTs in various pathologies, including AF, mainly in antioxidative effects [119,120]. Dapagliflozin, a SGLT2 inhibitor and sirtinol, a SIRT1 inhibitor, were used to investigate the role of the SGLT2-SIRT1 pathway in AF in a streptozotocin (STZ)-induced diabetes mellitus [121]. In addition, HL-1 cardiomyocytes were cultured under high-glucose (HG) conditions and treated with dapagliflozin, either in the presence or absence of sirtinol. In the rat model, dapagliflozin improved atrial fibrosis and reduced AF inducibility and duration-effects that were partially reversed by sirtinol. Therefore, dapagliflozin may alleviate cardiac fibrosis and atrial arrhythmia by modulating SIRT1. In HL-1 cells, dapagliflozin reduced apoptosis, restored autophagy and mitophagy, and enhanced calcium channel activity; however, these beneficial effects were reversed by sirtinol. Therefore, SIRT1 plays a protective role in diabetic cardiomyopathy and AF by reducing apoptosis, regulating autophagy and mitophagy, and modulating calcium channel activity.

The cGAS-STING pathway plays a key role in AF pathogenesis by detecting DNA damage, including in mitochondria, and activating genes that encode proteins involved in fibrosis and inflammation [122]. The activity of SIRTs, in particular SIRT2, SIRT3, and SIRTT4, was demonstrated to negatively regulate the cGAS-STING pathway [123,124,125,126]. Therefore, SIRTs are justified for further study in AF due to their involvement in the cGAS-STING pathway.

Aging is a main risk factor for AF due to cardiovascular aging and an age-related increase in comorbidity [127]. On the other hand, SIRTs are considered to have anti-aging potential [128]. Moreover, although the mitochondrial theory of aging is no longer the paradigm of aging, mitochondrial dysfunction remains central to the aging process [12].

In summary, SIRTs, beyond their basic deacetylase activity, also exhibit antioxidant properties that predispose them to influence oxidative stress-related mechanisms in AF development (Figure 3). This action of SIRTs is closely linked to mitochondrial health, especially since SIRT3, SIRT4, and SIRT5 are located within mitochondria and function there. Generally, SIRTs, particularly SIRT1, have shown potential to improve AF outcomes in studies involving both AF patients and animal models. The primary aspects of AF pathogenesis that SIRTs may influence include metabolic atrial remodeling, atrial fibrosis, age-related changes in necroptosis in the atrium, macroautophagy, and mitophagy, all of which are related to mitochondrial health and aging. However, a report suggests a RONS-driven mechanism of heart failure development, mediated by SIRT4, though that study requires further validation.

## 5. Conclusions and Perspectives

Atrial fibrillation is a common cardiac arrhythmia that can have serious health consequences, including stroke, heart failure, heart attack, pulmonary emboli, cognitive impairment, and dementia. It is treatable but generally considered incurable. Oxidative stress may be involved in AF development, similar to many other syndromes. A key question is whether there are specific aspects of oxidative stress in AF. Increased RONS production in AF is mainly due to damaged and dysfunctional mitochondria, supported by the high number of mitochondria in cardiomyocytes. Experimental studies show that RONS mediate AF induction. Several AF risk factors, such as heart failure and ischemic or nonischemic cardiomyopathy, are linked to RONS overproduction.

The main structural and functional changes in the myocardium during AF are structural and electrical remodeling, both of which are affected by mitochondrial RONS. Therefore, maintaining mitochondrial homeostasis through mtQC is essential to prevent AF, and impairments in this process may be a key element in AF development, making it a potential therapeutic target. Critical components of mtQC that can be impaired in AF include mitochondria-related calcium handling, energy production, mitochondria-related inflammation and fibrosis, and mitophagy. Numerous proteins and signaling pathways might contribute to these effects in AF, such as mitochondrial calcium transporters (MCU, NCLX, and RyR2); mitochondrial enzymes involved in OXPHOS; the cGAS-STING and MAPK14-AIFM2 pathways; NLRP3 inflammasome; PGC-1α, AGTR1, TGFB1, SDHA-D, *BAX*, *GLUL*, *MIF*, and *TLR4* genes; and PARKIN. Additional research is needed to clarify how these proteins and signaling pathways relate to one another and to the clinical presentation of AF. Moreover, further studies are required to explore how these factors interact, as mtQC involves an integrated set of cellular reactions responding to disturbances in mitochondrial homeostasis.

Sirtuins are notable in molecular pathology not only for their role as histone deacetylases but also due to their antioxidant properties. Additionally, three sirtuins function within mitochondria, and all are highly expressed in heart tissue, positioning them as natural candidates for linking oxidative stress and mitochondrial health in heart diseases.

Most research on sirtuins in human syndromes centers on SIRT1 and SIRT2. As noted, similar effects related to AF have been observed with different SIRTs, such as SIRT1 and SIRT3 in studies on atrial metabolic remodeling. This suggests that effects induced by a single SIRT may be mirrored by others. It is also important to recognize that correlations between observed effects and mRNA expression of sirtuin genes may not directly reflect the relationship at the protein level, as demonstrated by studies of SIRT1 and SIRT5 in age-related AF.

Current research indicates that some SIRTs, notably SIRT1, have beneficial effects in AF pathogenesis, primarily through their antioxidant actions. However, since SIRTs mainly act through deacetylation, it is necessary to investigate their influence on the expression of genes involved in AF, including *PITX2*, *BAX*, *GLUL*, *MIF*, and *TLR4*. This investigation could form part of a broader exploration of epigenetic regulation in AF development.

A negative regulation of the cGAS-STING pathway by SIRTs justifies further studies on SIRTs in AF pathogenesis, as the pathway may activate genes whose products are involved in fibrosis and inflammation.

A crucial question remains whether oxidative stress is a cause or a consequence of mitochondrial impairment, and whether it plays a role in AF pathogenesis. This is a complex issue, as oxidative stress can impair mitochondrial function, which in turn may increase RONS production and further aggravate oxidative damage.

In conclusion, disruptions in mitochondrial homeostasis may contribute to AF through multiple pathways, and sirtuins could help mitigate these disturbances. Therefore, they should be considered as part of a strategic approach to therapy for atrial fibrillation.

## Figures and Tables

**Figure 1 ijms-27-00175-f001:**
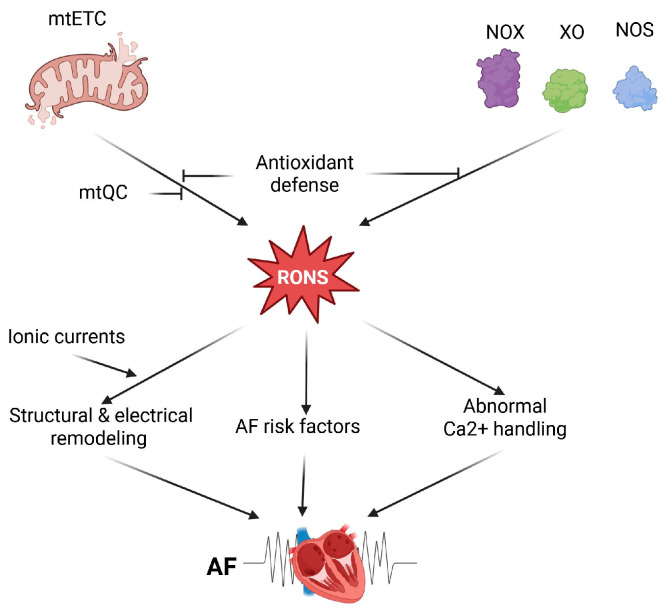
Oxidative stress in atrial fibrillation (AF). Oxidative stress in cardiomyocytes is linked to increased production of reactive oxygen and nitrogen species (RONS), mainly caused by impaired mitochondrial electron transport chain (mtETC) and activation of certain enzymes such as NADPH oxidases (NOX), xanthine oxidase (XO), and uncoupled nitric oxide synthase (NOS). The cellular antioxidant defense system regulates RONS levels, including those related to mitochondria, which are managed through mitochondrial quality control (mtQC). Elevated RONS levels, originating from NOX isoforms or mitochondria, may promote AF by causing abnormal Ca^2+^ handling in atrial myocytes, where levels are higher compared to ventricular myocytes. Several AF risk factors, including heart failure and ischemic or nonischemic cardiomyopathy, are associated with increased oxidative stress. Structural and electrical remodeling of the atrium is linked to AF and may be stimulated by RONS. Additionally, electrical remodeling is influenced by various redox-dependent ionic currents. Created in https://BioRender.com (accessed on 13 December 2025).

**Figure 2 ijms-27-00175-f002:**
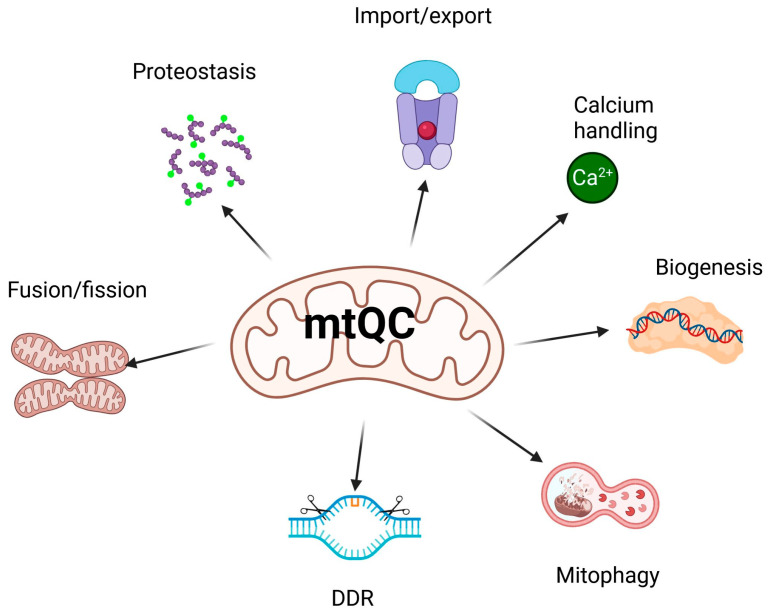
Main pathways of mitochondrial quality control (mtQC). Mitochondrial homeostasis is maintained by several coordinated mechanisms. These include mitochondrial biogenesis, which involves mtDNA replication and transcription, as well as fusion and fission processes. Protein homeostasis, through proteostasis and the import and export of proteins, also contributes. Additionally, mitophagy removes damaged mitochondria, while mechanisms of calcium handling and the mitochondrial DNA damage response maintain the integrity of the mitochondrial genome and ensure proper metabolism. Created in https://BioRender.com (accessed on 13 December 2025).

**Figure 3 ijms-27-00175-f003:**
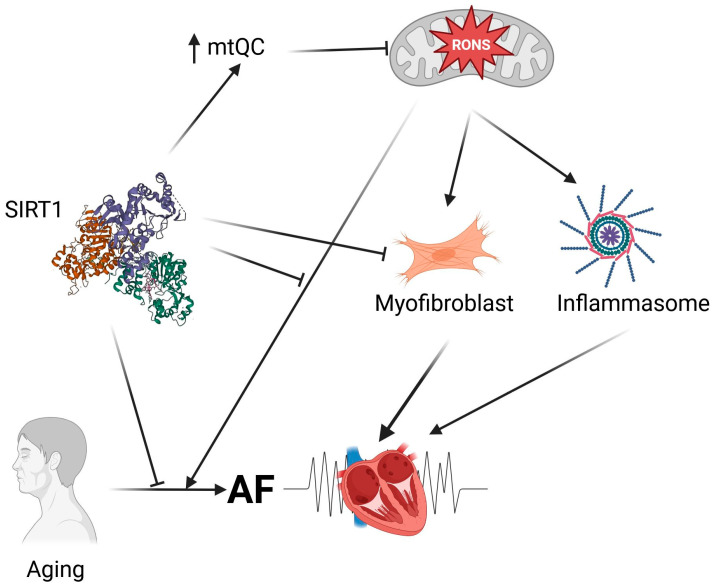
Sirtuins, represented here by SIRT1, may regulate mechanisms leading to atrial fibrillation (AF). SIRT1 may directly neutralize reactive oxygen and nitrogen species (RONS) produced by dysfunctional mitochondria. SIRT1 may also prevent overproduction of RONS by mitochondria by stimulating mitochondrial quality control (mtQC). Dysfunctional mitochondria produce molecules that stimulate signaling pathways resulting in fibrosis and inflammation, represented here by a myofibroblast and inflammasome, respectively. Aging, a risk factor for AF, can be influenced by both mitochondrial dysfunctions and sirtuins. Created in https://BioRender.com (accessed on 13 December 2025).

**Table 1 ijms-27-00175-t001:** Molecules that may exert antioxidant effects and ameliorate atrial fibrillation in experimental and clinical studies.

Molecule	Mechanism of RONS ^1^ Inhibition and AF Benefit	Reference(s)
Empagliflozin	↓ mitochondrial RONS → ↓ AF, fibrosis, conduction delay	[48]
Molecular hydrogen	↓ NOX4 → ↓ ROS, inflammation, fibrosis → ↓ AF	[41]
Licorice decoction	↓ NOX2 → ↑ SOD/CAT → ↓ ROS and TGFB1 pathway	[42]
Apocynin	Inhibits NOX2 → reduces atrial RONS and fibrosis	[37,43]
NOX2 inhibitors	Genetic/pharmacologic NOX2 targeting → restores electrophysiology	[37]
Atorvastatin	↓ NOX2-NADPH oxidase → effective in early/preventive AF	[17]
Vitamins C and E	General RONS scavengers → reduce AF recurrence	[17]
CoQ10, flavonoids, carotenoids	Broad antioxidant effects beneficial in CVD	[44]
NOS inhibitors	Target alternative RONS source in AF	[47]

^1^ Abbreviations: RONS, reactive oxygen and nitrogen species; AF, atrial fibrillation; NOX2/4, NADPH oxidase 2/4; SOD, superoxide dismutase; CAT, catalase, CoQ10, coenzyme Q10, CVD, cardiovascular disease; NOS, nitric oxide synthase.

## Data Availability

No new data were created or analyzed in this study. Data sharing is not applicable to this article.

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
