# Peer review of "Oxidative Stress, Mitochondrial Homeostasis, and Sirtuins in Atrial Fibrillation"

_ijms, 2025, doi:10.3390/ijms27010175_

Round 1
Reviewer 1 Report
Comments and Suggestions for Authors
This is a well-structured narrative review examining the interplay between oxidative stress, mitochondrial homeostasis, and sirtuins in atrial fibrillation pathogenesis. I congratulate the authors on their work. However, several areas require improvement to strengthen the scientific rigor and clinical relevance.
- The authors present oxidative stress as intimately linked to AF, but the causality direction remains unclear: "Oxidative DNA damage has recently been shown to be intimately linked to AF" - but is this cause or consequence? Secondly, the hydrogen peroxide perfusion studies (lines 133-134) in rats are cited as "remarkable examples" but represent acute pharmacological interventions that may not reflect chronic pathophysiology
- The section on mitochondrial calcium handling is compelling but relies heavily on one recent study. I suggest acknowledging the limited human data
- Before discussing about the NLRP3, I recommend incorporating a discussion of S100A8/A9 (calprotectin) as a mechanistic bridge between oxidative stress, inflammation, and AF pathogenesis. This would significantly strengthen your manuscript's integration of oxidative and inflammatory pathways. A recent comprehensive review has exhaustively addressed S100A8/A9 in cardiovascular disease with specific relevance to the mechanisms you discuss (DOI:10.3389/fimmu.2025.1630410). The authors can use this reference if they consider it useful.
- Strong claims about NLRP3 based largely on mouse models: Provide more detail on human studies (sample sizes, AF types, tissue sources) and discuss limitations of extrapolating from genetic mouse models
- The section that presents sirtuins as promising therapeutic targets has a minor issue, and I suggest addressing the SIRT4 paradox more thoroughly.
Reviewer 2 Report
Comments and Suggestions for Authors
In the manuscript: “Oxidative Stress, Mitochondrial Homeostasis, and Sirtuins in Atrial Fibrillation”, the authors examine the ... An interesting topic that contributes to knowledge in the area, but certain issues must be corrected.
Major revisions
- In the abstract and introduction, the objective of the manuscript must be mentioned. Moreover, authors must mention the gap the manuscript will fill in current knowledge, the results readers will find, and the possible conclusions readers will draw.
- The authors must specify the mechanisms by which reactive oxygen species (ROS) are produced and the antioxidants that degrade them to obtain the redox state, and then mention or define oxidative stress as a dysregulation of the redox state. Also, add mitochondria metabolism associated with ROS production.
- Authors must describe all the mechanisms they refer to and not merely state that they are reviewed by other authors, providing the references; the reader should be able to understand all the mechanisms without jumping between reviews.
- Many statements are not referenced, for example: “Dysfunctional mtQC was associated with numerous cardiovascular diseases, including ischemia-reperfusion, atherosclerosis, heart failure, cardiac hypertrophy, hypertension, diabetic and genetic cardiomyopathies, and Kawasaki disease.” Therefore, the authors should properly reference each point addressed in their review.
- The authors must expand on the description of the cited works; for example, “A recent study on material derived from AF patients confirmed the importance of impaired atrial mitochondrial calcium handling in the pathogenesis of AF.” In the above statement, the authors do not specify how many patients participated in the study, what type of results they obtained, or whether the results were significant.
- Overall, the review lacks a thorough analysis of the works examined. Furthermore, the authors should take a position on the reviewed works, indicating whether the findings are based on other research or require further studies to reach conclusions.
- The authors should acknowledge the limitations of their review and describe possible approaches to addressing ROS and mitochondrial metabolism in this disease.
- Authors cannot draw conclusions based solely on a single review. For example, “A retrospective review of patients found a lower AF burden among those taking ezetimibe. The authors concluded that mitochondrial Ca2+ uptake was impaired in atrial myocytes with AF, likely due to altered spatial organization between the retinopathy of prematurity (ROP) and mitochondria caused by microtubule destabilization. Improving mitochondrial Ca2+ uptake could serve as a protective strategy against arrhythmogenic events. These findings suggest that focusing on mitochondrial Ca2+ regulation could offer a novel therapeutic strategy for AF, particularly in persistent or refractory cases [52]”. The authors state that while the results of a review can lead to conclusions, they must review the studies included in that review and, after analyzing them, determine that mitochondrial Ca2+ uptake could serve as a protective strategy against arrhythmogenic events.
- The authors should explore additional antioxidants beyond resveratrol to reduce ROS in AF.
- Authors must use tables to present the different potential molecules for treating AF associated with ROS and mitochondrial homeostasis.
- Please organize and align all sections, as the first ones are very vague and the last ones are very long (reduce the information and avoid the textbook).
- Section 5, “Conclusions and Perspectives,” is lengthy. I suggest that the authors conclude each section after analyzing it and present only one conclusion in Section 5.
Round 2
Reviewer 1 Report
Comments and Suggestions for Authors
I congratulate the authors for their improvement; they have answered and have implemented all the suggestions.